# Invited perspectives: Fostering interoperability of data, models, communication and governance for disaster resilience through transdisciplinary knowledge co-production

Kai Schröter[1], Pia-Johanna Schweizer[2], Benedikt Gräler[3], Lydia Cumiskey[4], Sukaina Bharwani[5], Janne Parviainen[5], Chahan Kropf[6,17], Viktor Wattin Hakansson[6], Martin Drews[7], Tracy Irvine[8], Clarissa Dondi[9], Heiko Apel[10], Jana Löhrlein[11], Stefan Hochrainer-Stigler[12], Stefano Bagli[13], Levente Huszti[14], Christopher Genillard[15], Silvia Unguendoli[16], Fred Hattermann[18], and Max Steinhausen[1]

[1]Leichtweiß-Institute for Hydraulic Engineering and Water Resources, Division Hydrology and River Basin Management, Technische Universität Braunschweig, Beethovenstr. 51a, 38106 Braunschweig, Germany
[2]Research Institute for Sustainability - Helmholtz Centre Potsdam, Berliner Straße 130, 14467 Potsdam, Germany
[3]52°North Spatial Information Research GmbH, Martin-Luther-King-Weg 24, 48155 Münster, Germany
[4]MaREI: The SFI Research Centre for Energy, Climate and Marine; Environmental Research Institute, University College Cork, Beaufort building, Ringaskiddy, P43 C573 Cork, Ireland
[5]Stockholm Environment Institute, Oxford Eco Centre, Roger House, Osney Mead, OX2 0ES Oxford, United Kingdom
[6]Institute for Environmental Decisions, ETH Zürich, Universitätstrasse 16, 8092 Zürich, Switzerland
[7]Department of Technology, Management and Economics, Technical University of Denmark, Produktionstorvet B424, DK-2800, Kgs. Lyngby, Denmark
[8]Oasis Hub Ltd, 39,60 Barge Walk, SE10 0UG London, UK
[9]Agenzia regionale per la sicurezza territoriale e la protezione civile – Emilia Romagna, Viale Silvani 6, 40122 Bologna, Italy
[10]Section Hydrology, GFZ German Research Centre for Geosciences, Telegrafenberg, 14473 Potsdam, Germany
[11]Erftverband, Am Erftverband 6, 50126 Bergheim, Germany
[12]IIASA - International Institute for Applied Systems Analysis, Schlossplatz 1, 2361 Laxenburg, Austria
[13]GECOsistema srl, Piazza Malatesta 21, 47923 Rimini, Italy
[14]Zala Special Rescue, Épitök utja 5., 8900 Zalaegerszeg, Hungary
[15]Genillard&Co, Ismaninger Str. 102, 81672 Munich, Germany
[16]Hydro-Meteo-Climate Service of the Agency for Prevention, Environment and Energy of Emilia-Romagna (Arpae-SIMC), V.le Silvani 6, 40133 Bologna, Italy
[17]Federal Office of Meteorology and Climatology MeteoSwiss Operation Center 1, P.O. Box 257, 8058 Zürich, Switzerland
[18]Potsdam Institute for Climate Impact Research (PIK), Telegrafenberg, 14473 Potsdam, Germany

**Correspondence:** Kai Schröter (kai.schroeter@tu-bs.de)

**Abstract.** Despite considerable efforts and progress in increasing resilience to natural hazards, the adverse socio-economic impacts of extreme weather events continue to increase globally. As climate change progresses, disaster risk management needs alignment with adaptation measures. In this perspective paper, we discuss complications in disaster risk management that have manifested during recent events from an interoperability perspective. We argue that a lack of interoperability between data and models, information and communication, and governance are barriers to successful integrated disaster risk management and climate adaptation. On this basis, we take a detailed look at the challenges involved and suggest that trans-disciplinary knowledge co-production is key to promoting interoperability between these components. Finally, we outline a framework for

enabling knowledge co-production to enhance risk governance by improving ways of facilitating trans-disciplinary stakeholder engagement and draft a novel open-source federated data infrastructure, which allows stakeholders to consolidate and connect relevant data sources, models and information products.

## 1   Introduction

In July 2021, the low-pressure system Bernd smashed several rainfall records and caused catastrophic flooding and havoc in Central Europe, particularly in Germany and Belgium, with estimated losses exceeding EUR 30 billion and over 200 lives lost (Mohr et al., 2023). Only a few weeks later, temperatures around the Mediterranean Sea rose above 40 degrees Celsius and forest fires devastated extended areas. In the North of the Greek island Euboea, already charred landscapes without trees and other vegetation to hold water and soils suffered unexpected extreme precipitation, triggering floods and landslides, causing severe damage to local settlements and infrastructures. The series of hydro-climatic extreme events perpetuated in summer 2023 with flooding in the Italian Emilia Romagna region in May where after a long period of severe drought, heavy rainfall washed away houses and livelihoods (Arrighi and Domeneghetti, 2023), with extended periods of extreme heat stress in particular in Southern Europe, and with the record-breaking rainfall deposited by Storm Daniel which caused for instance catastrophic flooding in Libya in September. In all these examples and many others worldwide, such as the Henan floods in China (Hsu et al., 2023) or the Pakistan flood 2022 (Mallapaty, 2022), the impacts on local communities have gone beyond what has been previously observed. Besides the impacts of climate change, which alter the frequency and intensity of such events, other factors related to disaster risk management play a role when researching the causes of disasters and disasters avoided.

A closer inspection of flood risk management before and during the flood in July 2021 in central Europe reveals deficiencies in communication between actors, in information exchange between simulation models and users, and in institutional mechanisms for managing the risk and impacts. For instance, this applies to disseminating flood forecasts and meaningful warnings for the timely implementation of emergency measures (Thieken et al., 2023; Fekete and Sandholz, 2021; Mohr et al., 2023). Further, lacking awareness of the actual flood hazard and disregarding information about historic floods in extreme value estimation for defining low-probability flood scenarios (Vorogushyn et al., 2022) meant that the population was largely unprepared and emergency response organisations were overwhelmed. In the absence of historic flood observations or due to a lack of local flood experiences, expanding the view to hydrologically similar catchments is useful for learning about the possible magnitude of extreme floods and consequences from other regions (Kreibich et al., 2017; Bertola et al., 2023). However, anticipating local extremes using information from distinct places in Europe requires access to comprehensive data across regions and countries, which is not straightforward (Bertola et al., 2023). Whilst learning from other regions facing similar risks is possible, a lack of 'lived experience' and the uncertainty of the true impacts of the many interacting factors at play in different locales still prove

challenging to act upon. In addition, the uptake of such information among decision-makers and stakeholders is often hindered by wishful thinking biases (Ommer et al., 2024).

Despite these shortcomings, careful implementation of natural hazard risk management have demonstrated the capability to successfully mitigate impacts, even from unprecedented or surprising hazard events. A recent analysis of flood and drought-paired events pinpoints the reorganization of early warning systems and emergency response, improved collaboration between actors and the integration and exchange of data with enhanced accessibility of information as key factors (Kreibich et al., 2022, 2023). However, it is not enough to analyse and understand the dynamics of disaster risk in terms of what went

wrong and what worked well within the disaster risk management cycle (Schröter et al., 2018). It is uncontested that learning from thorough ex-post-event analyses is a fundamental step but must be turned into actionable recommendations to inform forward-looking risk management decisions, e.g. to build back better (Keating et al., 2016). Furthermore, some measures can be maladaptive, adding complexity through unexpected direct and indirect effects, e.g. the building of protection schemes can lead to increased settlement in flood-prone areas, thus increasing vulnerability (Kates et al., 2006; Burby, 2006; Haer et al.,

2020; Simpson et al., 2021). Management decisions need to be informed by an understanding of such interdependencies and the potential for maladaptation (Schipper, 2022).

    Owing to changes in climate, vulnerability and exposure (Merz et al., 2021; Steinhausen et al., 2022) disaster risk management is a continuous task and is closely intertwined with climate change adaptation. For instance, planners and decision-makers need to update design values to account for future changes and adapt e.g. structural defences and flood control systems or im-

plement nature-based solutions. The projection of extreme precipitation and other hydrological variables to future climate conditions for future flood risk assessment and climate adaptation are subject to research (Byun and Hamlet, 2020; Hattermann et al., 2018) but not anchored in planning processes. In some cases, safety margins for design values are applied to take into account the possible impacts of climate change. However, these are often not rooted in evidence-based analyses but are rather subjective and reflect the decision-makers' attitude to risk. Therefore, information on local climate change impacts on hazard

variables such as rainfall intensity and volume, prolonged dry spells or higher extreme temperatures needs to be produced, but importantly also needs to be accessible and embedded in practical planning processes in a structured and transparent way.

    Responsibilities for planning, implementation and management are distributed across administrative offices which complicates and impedes the exchange of information and communication. In addition, the communication of climate and disaster risks to the public is typically one-way and homogeneous but can be enhanced through two-way dialogues that identify, engage

and consult with specific stakeholders to develop tailored communications (requiring detailed analysis of the composition of different actors within an audience) and three-way participation where communication becomes a collective and continuous process of knowledge production between citizens, science and decision-makers (Stewart, 2024), e.g. using methods such as art- and citizen science, interactive games, role plays etc. Key differences and examples of the continuum between one-way, two-way, and three-way communications are provided by (Stewart, 2024). Moving from product to process, for instance, the

Tandem framework (Daniels et al., 2020) was applied in a southern African urban context addressing adaptation and disaster risk challenges in peri-urban areas using a trans-disciplinary 'Learning Lab' approach and 'embedded researchers' to bring stakeholders together to identify and prioritize challenges and co-create solutions and creating long-lasting relationships, which

support ongoing networks such as the public-private multi-stakeholder partnership, e.g. the Lusaka Water Security Initiative (LuWSi). Recent applications (Bharwani et al., 2024) diversified Tandem's use in different socioeconomic settings and decision domains. A rural Indonesian community of smallholder coffee and cacao farmers co-created weather forecasts with the national meteorological office to tailor farmer field school curricula with local ecological knowledge, concepts and terminology. In Sweden urban planners, meteorological scenario modellers, hydro-climatologists and city officials co-explored compound events related to flooding (cloudburst events and spring floods) as well as heatwave scenarios to inform the city's 2024 Stockholm's Environmental Programme (2020-2023) and the Climate Adaptation Action Plan (2022-2025). In Colombia, a participatory group, the river basin council, including representatives from local and regional communities and institutions addressing water scarcity and inequitable access (farmers, municipalities, NGOs, indigenous populations and the private sector), co-designed a graphical web tool interface that translated hydro-meteorological data into accessible, relevant and usable information and language for basin planning, which continues to be used today. All of these processes enhanced information interoperability, as well as the capacity and confidence of stakeholders to work with and recognize the limits of climate information (Bharwani et al., 2024).

The above non-exhaustive selection of examples - with a predominant focus on floods - indicates a pattern of complications between the phases of the disaster risk management cycle, between the various institutions and actors involved, between the data, methods and tools deployed, and, on a higher level, between the planning domains for disaster risk management (DRM) and climate change adaptation (CCA). We argue that these complications emerge from barriers or gaps within the practical implementation of DRM and CCA at different levels as illustrated in **Fig.1**. DRM and CCA involve different governance levels at (inter-) national, regional and local scales. The phases of the risk management cycle are carried out under the direction of various planning departments with different responsibilities, often with the involvement of consultants and other actors such as emergency management, insurance companies, etc. Data and models are usually applied for dedicated tasks in a sectoral approach and are often agnostic of parallel or subsequent activities. One example is the production of flood hazard and risk maps during the implementation of the European Floods Directive (2007/60/EC, 2007) with diverging definitions regarding the extreme flood scenario, which leads to inconsistencies in hazard and risk information across federal state or national borders and eventually confuses decision making for transboundary flood risk management (Thieken et al., 2016).

We attribute the emergence of gaps to a lack of interoperability within governance structures and processes, communication and knowledge exchange between responsible institutions and actors in the different phases of the risk management cycle – prevention, preparedness, response and recovery – and a lack of interoperability regarding data, models and tools which support the many different tasks and decisions in disaster risk management. In general terms, interoperability refers to the ability of a system to work together with other systems or pieces of equipment or products [Cambridge Dictionary]. From a technical viewpoint, syntactic interoperability means the ability of two systems to communicate with each other and cross-domain interoperability refers to multiple organizations working together and exchanging information. Concerning data, for example, the Infrastructure for Spatial Information in the European Community (INSPIRE) directive defines interoperability as "the possibility for spatial data sets to be combined, and for services to interact, without repetitive manual intervention, in such a way that the result is coherent and the added value of the datasets and services is enhanced".

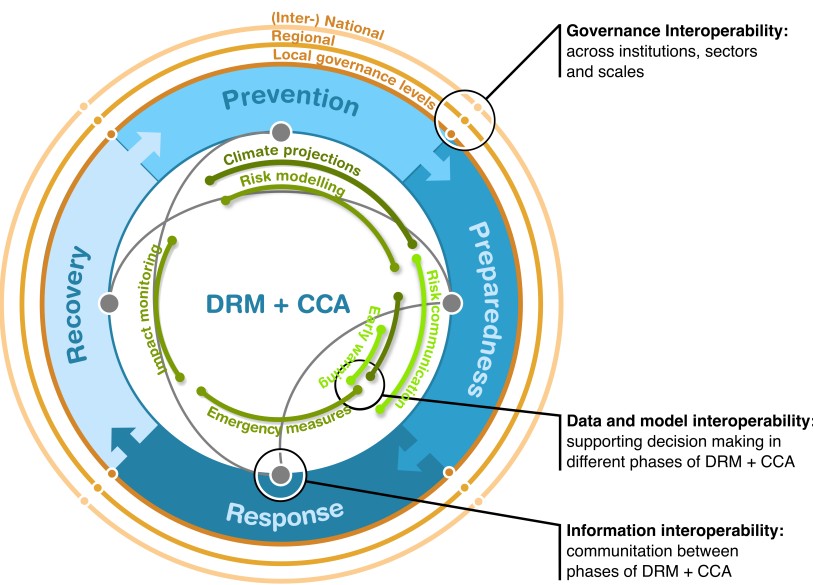

**Figure 1.** Disaster risk management and climate change adaptation involves different governance levels at (inter-) national, regional and local scales. The phases of the risk management cycle are carried out under the direction of various planning departments with different responsibilities, often with the involvement of consultants and other actors such as emergency management, insurance companies, etc. Data and tools are usually applied for dedicated tasks in a sectoral approach and are often agnostic of parallel or subsequent activities. The illustration exemplifies selected gaps (black circles) to evince the critical role of interoperability at different system levels.

We suggest, that the concept of interoperability can be applied to DRM and CCA in the sense that the exchange between different systems is necessary as a prerequisite for informed and efficient decision-making which ultimately improves disaster resilience. In these domains, systems can be different actors, responsible institutions, and areas of responsibility in the phases of the risk management cycle. Actors involved are for instance, civil protection, firefighters, healthcare services, municipalities and non-profit organizations among others for whom the exchange of information and communication is essential for cooperation e.g. in emergency response (Migliorini et al., 2019). Systems also refer to dedicated models, tools or specific databases or providers, which are used in the various DRM and CCA tasks for decision support. Using vast amounts of heterogeneous data for efficient and informed decision support presents many challenges regarding accessibility, comparability, quality, organisation and dissemination of temporal and spatial data for natural hazards (Tomas et al., 2015). These various systems, i.e. data and models, information and communication, and governance should be understood as parts of an overall DRM and CCA system, which requires coordination for the interaction between the individual components (Vercruysse et al., 2019).

Both in higher-level DRM and CCA systems, and in these three systems, we see a variety of interoperability challenges. For data and models, these relate to technical aspects of data accessibility, security, privacy and licensing as well as longevity, information about uncertainty, transparency and contextualisation, e.g. in terms of metadata. For information and communi-

cation, interoperability challenges pertain to the sector-specific understanding of information including underlying encoded (subjective) assumptions, the exchange of knowledge across disciplines and uptake in decision-making, as well as oversights, replication and redundancy of communication and information use, due to information silos. Interoperability challenges emerge from governance, financial and legal mechanisms and processes including political will and disconnected short-term financing, ambiguity in complex decision processes with wide implications as well as harvesting the best available knowledge from various knowledge repositories.

In this perspective paper, we discuss interoperability challenges for DRM and CCA by taking a detailed look at data and models, information and communication, and governance systems (Chapter 2). On this basis, we propose recommendations for overcoming these challenges (Chapter 3) based on research and development work carried out in the inter- and transdisciplinary EU innovation project DIRECTED which aims to reduce vulnerability to extreme weather events and foster disaster-resilient societies by promoting interoperability between DRM and CCA. While the learning from real-world-labs within the DIRECTED project is based on the specific conditions given in these European settings, the methods proposed will be applicable in other geographical and cultural contexts. We summarise our perspectives on interoperability for disaster resilience through transdisciplinary knowledge co-production (Chapter 4).

## 2  Challenges

The three systems - data and models, information and communication, and governance - each pose different and specific challenges to interoperability, and are described in the following sections and illustrated using examples.

### 2.1  Challenge 1: Data and model interoperability

We begin our exploration with the foundational aspects of, in particular, digital data — specifically, issues of format compatibility, standards, and data availability and accessibility. Moving forward, we discuss the challenges related to the coupling of models, such as using data outputs from one model as inputs for another, acknowledging that this can result in a cascade of uncertainties due to model imperfections and biases. We highlight the necessity of making these biases transparent across models and data sources. Finally, we address the challenges related to the usage and interpretation of data and models, emphasizing the critical importance of contextual knowledge and the imperative of transferring expertise across various domains.

Data is the foundation of any digital information product. Consequently, the accessibility and quality of data have a direct impact on the possible quality of the information product and the decisions taken. The principles of FAIR data (Findable, Accessible, Interoperable and Reusable) were published in 2016 (Wilkinson et al., 2016) and promoted since, but it has not yet become standard. Even if data can be found and accessed in an interoperable way, reusing it is often not straightforward. Different (open) standards exist across, but also inside, various domains. In DIRECTED, most data sets have a spatial and temporal reference and follow standards developed by the Open Geospatial Consortium (OGC, 2024) that are largely accepted. However, the OGC also offers more than 30 geospatial data and information exchange standards. Whether the desired data set

and the tool to further analyse and process the data are compatible is not guaranteed and often intermediate steps to further transform and map the data are necessary.

The availability and management of data are challenged not only by the technical aspects of interoperability but also by legal and operational constraints. Licensing terms can restrict the use and sharing of data, since, for instance, scientific users might not have a legal team to handle subtleties of non-open licenses (e.g., CC BY-NC-SA), non-commercial users might not have the financial means to pay for expensive data licenses, or commercial users might not want their derived products to have open licenses (e.g., CC BY-SA). Furthermore, privacy regulations, such as those mandated by the General Data Protection

Regulation (Regulation (EU) 2016/679), while necessary to protect basic human rights, impose additional layers of complexity, often requiring data anonymization. For instance, simply revealing the total number of minors in a vulnerable region, while potentially valuable for risk assessment models, could also be exploited by Human trafficants, and thus requires extra care and consideration before sharing. This can inadvertently increase the complexity of the data or the models used. Another critical interoperability issue is the longevity and upkeep of necessary external data and models, particularly those developed through

open-source projects initially supported by research funding. When the funding dries up, these projects often face the risk of becoming 'dead data', i.e. inaccessible, outdated, or irrelevant. The uncertainty surrounding the longevity of open data and models also stems from a lack of explicit commitments from the providers. The creation of open models and datasets requires not just a dedication to openness but also a significant investment of time. Often, these efforts are sidelined in favour of short-term efficiency, constrained budgets, or individual career objectives. This compromises interoperability and the potential for

long-term benefits, introducing a risk for users in deciding whether or not to rely on open data models. While this results in a data challenge, it is ultimately a governance and cultural challenge. Besides the technical compatibility, metadata is key to meaningfully using the acquired data and modelling results in subsequent processing steps. Information about the semantics of the data, its origin and provenance and also normative assumptions need to be understood.

    In the subsequent processing through models and chains of models, their final and intermediate outputs are again data and

175 the same challenges apply. However, if additional transformations of one model's output become necessary to be used as another model's input, this stresses the importance of data interoperability. Furthermore, modelling typically implies certain assumptions on the inputs and the outputs, as well as modelling biases (Wang et al., 2014) and uncertainties (Pianosi et al., 2016; Kropf et al., 2022). Managing and communicating uncertainties - a result of data quality, model assumptions, or inherent biases - becomes increasingly complex as networks of models expand. Quality control, harmonization and contextualization

are essential at each of the data transfer points between the models to prevent the magnification of errors and uncertainties along the model chain. Thus, additional information needs to be added to the data and treated sensibly. While the UncertWeb (Bastin et al., 2013) project and other initiatives developed some technical tools and standards, their application in practice are still rare.

    Decision-making biases and uncertainties don't just arise from model imperfections and measurement errors but also from

185 deeply embedded normative assumptions that shape model outputs. These assumptions may influence adaptation planning substantially, e.g., local and global economic models may respond sharply to changes in the underlying assumptions including growth and development projections (Halsnæs et al., 2015; Buhaug and Vestby, 2019). Simple pathway-based models includ-

ing the Shared Socio-economic Pathways (SSPs) (O'Neill et al., 2017) may streamline conceptual understanding and future planning, but they can also gloss over the intricacies of uncertainty modelling in real-world settings. This emphasizes that transparency and contextualization in how data and models are constructed becomes crucial, which can be even more difficult due to the proprietary nature of commercial data and model providers (Arribas et al., 2022).

Furthermore, even if all data is interoperable and well documented by metadata, it remains challenging to use it in practice as users can rarely be experts in all the fields required to judge the fitness for purpose adequately. The crux of data and model interoperability rests on finding the equilibrium between model complexity and the granularity of input and output, harmonized with stakeholder-specific preferences as well as collective contextual understanding and discernment. An overabundance of detail in data and metadata can obscure vital insights, and an overly intricate and complex model at one stage may be suboptimal to the performance of subsequent applications. The objective is to reflect on important aspects for different stakeholders and to supply the appropriate amount and level of data and contextual knowledge sharing necessary to inform the next stages in modelling or decision-making processes. While several technical solutions such as the definition of data standards are pre-requisite for this objective, direct exchange between experts of connected models and data is often the only way to achieve maximum interoperability.

Within DRM and CCA, a synergistic interplay exists between the application of local expertise and the analytical prowess of data-driven models, each contributing its unique insights to enhance risk assessment. Local experts need to harness weather model warnings to guide immediate decisions, like emergency resource allocation, or to shape strategies for future adaptation to unprecedented risk levels. However, these local judgments often hinge on historical data (Šakić Trogrlić et al., 2019), which may not account for new or evolving risk patterns. Climate models, on the other hand, are less constrained by historical data and more reliant on dynamic modelling, essential for proactive disaster response and adaptation planning. Due to their forward-looking nature, uncertainties and biases, subtle or overt, human oversight in interpretation is a necessity.

Hence facilitating knowledge sharing and iterative and reflexive communication among users, data providers, modellers, policymakers and even the general public is essential. These interactions are pivotal not only for systematically mitigating errors that propagate through data and model chains but also for identifying and addressing uncertainties and biases that could affect decision-making based on the data, model outputs and their interpretation. By providing insights into subjective knowledge and inherent biases, they highlight the necessity for contextual understanding and nuanced model interpretation.

### 2.2 Challenge 2: Information and Communication interoperability

Translating data and knowledge into action is often missing due to a lack of understanding of user needs and the integration of different stakeholder perspectives and aspects, knowledge and disciplines. This can compound the lack of collaboration between data providers themselves as well as with users and other stakeholders and lead to a dismissal of values and norms that inform decision-making and therefore affect the uptake and application of data and information as they may not be fit for both purpose and context. These challenges filter down to citizens and the public and require inclusive dialogue, knowledge exchange and effective translation to result in coordinated action.

Translating climate risk information and research into policy and action has been severely lacking and remains challenging (Klein and Juhola, 2014; Brasseur and Gallardo, 2016). Often, climate information and data remain overly technical or are developed in isolation from user needs, which thus reduces its usefulness for, and uptake by, decision-makers (Lemos et al., 2012). Similar challenges limit the application of information on complex risks to policy (Sillmann et al., 2022), not to mention the tendency of actors and research to produce knowledge in disciplinary silos that constitute tunnel vision. Due to the difficulties concerning the availability and interoperability of relevant data sets (Challenge 1), users face additional challenges in understanding this information well enough to be able to use it and connect to the producers as well as other experts. Hence, the taxonomies and general approaches implemented by different models may not be consistent across institutions or scientific domains - or even across models from the same field (Barrot et al., 2020). As such, technical data interoperability must be accompanied by efforts to support the interoperability of the information and communication channels and approaches seeking to address these issues (cf. Challenge 1 in section 2.1). In addition, the underpinning assumptions and values embedded in data systems represent a challenge often left undiscussed. After all, the selection of parameters for climate, hazard and risk modelling is also a subjective (and political) process, e.g. in terms of extreme event attribution and discourses surrounding the concept of 'loss and damage' (McNamara and Jackson, 2019; Olsson et al., 2022). For example, merely the effort to assess an event as 'extreme' involves the introduction of human value judgements into the scientific process, thus making it a subjective claim (Olsson et al., 2022). Values can also be found in conceptualizing mapping, and modelling losses and damages with extreme events; these tend to emphasize economic dimensions at the expense of impacts in terms of human well-being (McNamara and Jackson, 2019). In the wider global context, the trends in scientific discourses reflect (primarily Western) epistemological assumptions which continue to shape what is perceived as 'trusted' knowledge and solidify certain ways of knowing over others (Shawoo and Thornton, 2019; Funk and Guthadjaka, 2020). This means that not all knowledge and knowledge types (e.g. local or practitioner as well as scientific) are included as 'trusted' knowledge in a particular domain. Thus, and in alignment with the commitment to 'communication interoperability', there is a need to accommodate critical reflexivity regarding who has or should have the right, mandate, power and position to communicate, and how.

A related challenge for interoperability between actors is found between data providers and users, including decision-makers and planners. Considering that the efficacy of data is often limited by its isolation from user needs (Lemos et al., 2012), the absence of multi-scalar collaboration and communication continues to propagate the development of solutions and products - such as risk models and scenarios - that are not fit for management purpose and decision-making context. Meanwhile, knowledge silos and discrepancies between actors at different levels are likely to constitute oversights, replication and redundancy of communication and information use. Finally, issues of communication interoperability may extend beyond the circles of scientists, experts, modellers, and decision-makers to citizens and the broader public. Risk information is typically made accessible through one-way top-down dissemination from authorities to the broad public, e.g. hazard warnings/alerts or risk maps. This poses numerous challenges – especially in terms of early warnings. Even the most elaborate forecasting models and early warning systems will be rendered ineffective in the absence of a timely, clear and tailored communication (Lemos et al., 2012; Kreibich et al., 2017; Fakhruddin et al., 2020). One example is flood warnings, which only lead to a noticeable reduction in

damage if those warned know what to do, underlining the need for warnings to contain helpful and specific instructions for action (Kreibich et al., 2021).

## 2.3 Challenge 3: Governance interoperability

Previous sections outlined the challenges stemming from a lack of interoperability related to data and models as well as communication. Both challenges impact governance interoperability. Most prominently, challenges pertain to knowledge integration,
a reflection of the wider social implications of policies, and goal alignment of governance mechanisms across organisations. Governance broadly refers to the structures and processes "through which society and the economy are steered towards collectively negotiated objectives" (Ansell and Torfing, 2016). Risk governance more specifically includes the totality of actors, rules, conventions, processes and mechanisms concerned with how relevant risk information is collected, analysed and communicated, and how management decisions are taken (Aven et al., 2018). Risk governance especially draws attention to the
institutional structures and socio-political processes that influence collective activities in the context of decision-making on risk (Klinke and Renn, 2021). As such, risk governance is concerned with both analysing risk based on up-to-date information and the best available knowledge as well as decision-making on risk based on a reflection of the implications of those decisions might have (Florin and Bürkler, 2018).

First, the ideal of pooling all relevant information requires sharing of data, which comes with various challenges as outlined
above. In addition, harvesting the best available knowledge requires tapping into diverse knowledge repositories to garner information on hazards and risks and their further implications for DRM and CCA (Kelman et al., 2012; Mercer, 2012; Weichselgartner and Pigeon, 2015). Therefore, stakeholder and public engagement in risk governance are motivated by the realisation that these groups provide crucial information for assessment from diverse standpoints and perspectives, including scientific knowledge and other knowledge systems (Fischhoff, 1995). The 'operationalisation gap' between knowledge generation and
its uptake in DRM and CCA policy and practice has been well described within interdisciplinary research projects, e.g. the Enhancing synergies for disaster prevention in the European Union (ESPREssO) project addressed the integration of DRM and CCA (Zuccaro et al., 2020). Thus, a key governance challenge is streamlining the processes of information and knowledge sharing and synthesising among diverse stakeholders including knowledge producers and users. This challenge is highly intertwined with others discussed above, including siloed knowledge production and communication gaps.
Second, risk governance is concerned with taking decisions on risk which requires a reflection on the wider societal implications of policies and governance measures based on value judgements. In the context of DRM and CCA decisions, the underlying risk problems are rarely ever simple, i.e. characterised by simple cause-and-effect chains such as that smoking tobacco is linked to an increase in cancer risk. Rather, DRM and CCA decisions touch upon complex risk problems for which it is difficult to predict the performance of the system considered, based on the dynamic and highly complex interplay of its
constituent parts and intervening external factors (Aven and Renn, 2020). In addition, risk problems are associated with uncertainty if it is difficult to accurately predict not only the occurrence of events but also their potential consequences (Aven and Renn, 2020). Linking with the discussions above on data and information challenges, this uncertainty can be due to incomplete or invalid data and/or modelling inaccuracies. Furthermore, ambiguity arises if there are different views on relevance,

meaning and implications of the information for decision-making (interpretative ambiguity) (Aven and Renn, 2020). In addition, ambiguity arises about the underlying worldviews, norms and values that need to be protected and which priorities for selecting policies and governance measures follow from these considerations (normative ambiguity) (Aven and Renn, 2020). All of these risk issues are at play when considering DRM and CCA. As a consequence, the governance of DRM and CCA needs to consider the concerns and value judgements of relevant interest groups and the affected public. This requirement calls for discourse-based strategies of inclusion and participation that go beyond mere stakeholder consultation for assessing the implications of management options (Schweizer and Renn, 2019).

Third, as a consequence of the previous challenges, risk governance is concerned with streamlining financial and legal mechanisms as well as processes of interactions between actors and institutions. More specifically, disaster risk-related governance takes place in multi-faceted forms ranging from international environmental agreements and national legislature to activities without strict legal character, such as climate adaptation or disaster risk management strategies, action plans, non-binding programs, and mechanisms. These multiple forms of governance horizontally and vertically link state and non-state actors in a complex, multi-level governance configuration. However, actor responsibilities for reducing disaster risk and adapting to climate change are fragmented across multiple institutions, sectors and levels, thus requiring mechanisms to enable integration and interconnectedness across governance systems (Keast et al., 2007; Gilissen et al., 2015; Cumiskey et al., 2019). Further governance challenges hindering DRM and CCA integration in practice include a lack of capacity at different levels but especially at the local level, and a lack of political will, combined with disconnected and short-term financing arrangements (Birkmann and Von Teichman, 2010; Dias et al., 2020). Although the need for governance coherence, alignment and integration across DRM and CCA enabling policy frameworks and practice have been identified (Mysiak et al., 2018; Schweizer and Renn, 2019; Medway et al., 2021; Adams et al., 2020; Leitner et al., 2020), good practice reports on cross-border and -sectoral collaboration amongst multiple stakeholders in multi-level settings are challenging to align and facilitate due to fragmentation and silos, lack of agency and capacity, as well as diverging jurisdiction and accountability. Despite continued calls for more coherence between DRM and CCA policy and practice, good practice examples of methods, policies and practices capturing synergies are limited (Mysiak et al., 2018; Booth et al., 2020; Deubelli and Mechler, 2021). This governance challenge aligns with the challenge for data/information providers to produce fit-for-purpose or tailored DRM and CCA data, information and tools that meet policymakers' needs.

## 3 Propositions

Given the complex challenges of interoperability in data and models, information and communication, and governance, we propose that inclusive transdisciplinary knowledge co-production processes are the key to fostering interoperability between these systems to manage complex and interconnected climate and disaster risks. We define knowledge co-production as "transdisciplinary engagement processes through which public, private and civil society actors produce and integrate knowledge toward resilience and sustainability", following Norström et al. (2020) and Daniels et al. (2020).

The challenges discussed in Section 2 can be traced to disciplinary silos, lack of collaboration and the fractured landscapes of academia, decision-making, practice, financing and policy. As such, it is evident that interoperability for DRM and CCA hinge upon the cultivation of relationships and connections which can support collaborative action, knowledge integration, and joint learning. Knowledge co-production holds the potential to enable transdisciplinary collaboration for more effective management of complexity, environmental change, and risk (Cosens et al., 2021; Norström et al., 2020; Polk, 2015; Mees et al., 2017; Brugnach and Özerol, 2019). For the three above-mentioned systems, this is possible in the following ways.

## 3.1 Sub-system interoperability

*Data and models:* To respond to data interoperability needs, for instance, co-production may support the demand-led tailoring, preparation and provision of 'usable' as opposed to just 'useful' information (Lemos et al., 2012). For instance, the Copernicus Climate Data Store provides various Climate Services that are 'useable' through a web portal or the open APIs. However, its usefulness depends on the actual user. A farmer in the field for instance, will find those Climate Services hardly useful without additional processing and information derivation tailored to his needs and day-to-day questions. Connecting users of data with data providers, with the aim to identify an equilibrium between model complexity and the granularity of outputs for planning and decision-making will foster an increased use and uptake of climate and risk information. Indeed, establishing avenues for transdisciplinary collaboration and communication between producers and users of data is essential to develop processes (e.g. sustained relationships and connections) as well as information products that are fit for purpose, and thus maximize their impact. For example, the Flood Resilience Measurement for Communities (FRMC) tool not only quantitatively assesses different sources of resilience against flood risks on the community level but also highlights strengths and weaknesses in community resilience using different perspectives that can be visualized, arranged and displayed flexibly according to the user needs (Zurich Flood Resilience Alliance 2019; https://zcralliance.org/resources/item/the-flood-resilience-measurement-for-communities-frmc/).

Strategic co-production may enable knowledge sharing and the reflexive communication between data providers, modellers and policymakers to assess and identify potential errors and uncertainties embedded in data and model chains, but also to evaluate biases underpinning modelling assumptions.

A possible way to embed this in existing workflows is to build standardized and simplified data and model pipelines, which encapsulate some of the complexity and allow for the easy running of separate models addressing the same question. This allows comparing model outputs of different approaches, which can support building trust in the models to be fit for the intended purpose.

While it should be acknowledged that running different models does not inherently enhance trust in the models, in particular when outputs are highly uncertain and variable, using several distinct modelling approaches building up to shared evidence and a common understanding of the phenomenon of interest is one of the main ways for scientists to build trust in models that are only hardly verifiable (Taylor et al., 2012; Merz et al., 2024). For instance, climate models may yield very large variances in global temperature outputs for the same emission trajectories. Confidence in the climate models is established by pooling the information from several distinct models. This confidence was built thanks to a large-scale interoperability effort

led by the CMIP team. This type of multi-model angle is not only required on the physical modelling side, where it is already common throughout the entire chain of information, but ideally for all models to best characterise uncertainties. However, this often shows how large they are, and is often difficult for stakeholders to fully understand. More importantly, disclosure of uncertainties does not inherently increase trust and credibility in risk analyses (Doyle et al., 2019), and thus information about uncertainty should be embedded in the co-production process and tailored to the specific audience, considering their perspectives, technical knowledge and concerns (Merz et al., 2024). Crucially, to bridge this gap, there arises a need to build trust among modelers, stakeholders and decision-makers. Effective communication plays a pivotal role in establishing trust in DRM and CCA decisions. The proposed knowledge co-production process, i.e. connecting modellers, data providers and end users, promotes discussions regarding different modelling approaches and explores user needs vis-à-vis available information in a non-hierarchical manner (Daniels et al., 2020). This is expected to increase the usability and accessibility of information by clarifying potential errors, uncertainties and underpinning assumptions embedded in each model for users, to align available information with the needs of planners and decision-makers. In other words, the co-production process is an ongoing negotiation between needs and what models can provide, which, insofar as uncertainty is accounted for, also generates trust in data through continuous and transdisciplinary engagement with it (Daniels et al., 2020).

Although knowledge co-production cannot necessarily address issues related to open data and the longevity of data systems, it represents a starting point for open and transdisciplinary collaboration towards addressing the gaps between available information and its use. However, co-produced information products tailored to user needs to maximise their usefulness fosters interest and motivates longevity.

*Information and communication:* Alongside the challenges of data interoperability, knowledge co-production can also contribute to addressing gaps in information exchange and communication between stakeholders. For example, collaborative and interactive methods can be leveraged to deconstruct disciplinary silos, with a commitment to unpacking, explaining, and interpreting field-specific or otherwise highly contextual concepts with actors in transdisciplinary settings as an attempt to support knowledge integration in planning for disaster and climate resilience (Daniels et al., 2020; André et al., 2021; Bharwani et al., 2023; Jack et al., 2020). Creative and interdisciplinary approaches to communicating complex and uncertain information can help embrace the audiences' characteristics and foster emotional responses (Stewart, 2024; Balog-Way et al., 2020). Interactivity is one of the key principles of knowledge co-production (Norström et al., 2020). Creative approaches can include interactive games to co-explore issues, language and terminology (Daniels et al., 2020) and participatory arts-based methods to support three-way communication between decision-makers, scientists and communities (Stewart, 2024). These aim to clarify and unpack complex and sometimes field-specific knowledges so that they become easily accessible for all stakeholder groups; for instance, with the help of interactive and game-based methods such as tabletop exercises, serious games, or "future-casting" and visioning. In practical terms, modelling, information and communication and governance systems will benefit from a commonly understood language to strengthen interoperability. Co-production can be leveraged in the efforts to build harmonized language regarding shared challenges and potential solutions. More importantly, it can build trust and the formation of new relationships (Fledderus, 2018; Norström et al., 2020), thus constituting the foundation upon which communication interoperability can be built. It also has the potential to alter and shift the dynamics of power between stakeholders involved – which,

beyond being a normative agenda – is important for building new coalitions and identifying transformative pathways toward change beyond a status quo (Cosens et al., 2021; Wyborn et al., 2019).

*Governance:* This latter potential also links knowledge co-production and the wider challenges of risk governance. For instance, communication gaps as discussed above are often underpinned by siloed knowledge production, lack of formal processes and institutions or knowledge-sharing mechanisms between stakeholders that could enable collaboration. Facing
this complexity, co-production can be leveraged to "co-produce governance", by expanding and opening the landscape of governance to actors who are not traditionally considered a part of it (Cosens et al., 2021). It may also provide an opportunity to reflect on the wider socio-economic and political context and values underpinning risk management decisions. As pointed out by Wyborn et al. (2019), long-term, iterative and inclusive co-production processes can help develop consensus and trust among stakeholders facing potentially controversial or sensitive topics. Co-production with diverse groups of actors requires
tailored governance arrangements that facilitate collaboration, and knowledge integration and encourage innovation across science, policy and practice. This requires opening up risk governance to deliberative and participatory processes of stakeholder engagement and public participation, which become a source of legitimacy in governing risk and incorporating extended peer, stakeholder, and public communities (Klinke and Renn, 2021). Inclusive risk governance assumes that stakeholders make valid contributions to the process of risk governance and that mutual communication and exchange of ideas, assessments and
evaluations improve disaster risk management by providing substantial knowledge rather than impeding the decision-making process or compromising the quality of scientific input and the legitimacy of legal requirements (Renn and Schweizer, 2009; Schweizer and Renn, 2019). Evidence-based risk analysis provides relevant scientific facts needed for risk characterisation and risk management. Governance of disaster risks consists of many dimensions, e.g. the rule structure, the distribution of resources, value orientations and cultural settings, as well as attitudes and beliefs. Thus, the inclusion of stakeholders and and
civil society is expected to provide a normative yardstick for evaluating characterisations and the management options available (Schweizer and Renn, 2019; Schweizer, 2021).

In addition to these general requirements of risk governance that operate via inclusive and transdisciplinary knowledge co-production, specific roles and capacities are needed to facilitate such transboundary activities supporting transdisciplinary co-production. These roles support trans-disciplinary knowledge co-production through their capacity to network, communicate,
facilitate and convene multiple diverse stakeholders, navigate their different goals/values and synthesise and integrate/broker knowledge to influence policy or practice. This entails identifying champions and facilitators capable and interested in bridging connections between disciplines and knowledges involved (Bharwani et al., 2024), in consideration of the skills and capacities required for them to enable co-production as described above. These roles are recognised through a range of terms emerging from practice and policy e.g. boundary-spanning roles (Williams, 2011), collaboration champions (Crosby and Bryson, 2010),
policy entrepreneurs (Meijerink and Huitema, 2010) and academia/ science, e.g. integration experts (Hoffmann et al., 2022) and knowledge brokers (Cvitanovic et al., 2016; Meyer, 2010). Such facilitators of co-production can work within existing collaborative structures or partnerships e.g. social learning within Regional Flood and Coastal Committees in England (Benson et al., 2016) or developing new place- or theme-based 'laboratory' arrangements to embrace collaborative learning through experimentation e.g. Learning Labs for climate information co-production in Lusaka, Zambia (Daniels et al., 2020), Amsterdam

Crisis Resilience Living Lab (Boersma et al., 2022), and Baden-Württemberg Labs for transdisciplinary sustainability research (Bergmann et al., 2021).

## 3.2 Innovations for interoperability

Considering the above-mentioned recommendations, the DIRECTED project proposes two interconnected innovations: The **Risk-Tandem framework** and the **Data Fabric**. The DIRECTED project with its trans-disciplinary consortium of researchers,
practitioners and industry partners has begun its work in late 2022 and is scheduled to conclude in September 2026. The project's main objective is to strengthen the disaster resilience of societies by overcoming existing silos in DRM and CCA. DIRECTED aims to enhance interoperability between modelling tools used in DRM and CCA to improve communication and governance processes. These innovations for interoperability are co-developed and implemented in an iterative approach with local project partners and stakeholder groups in four European Real Word Labs in Denmark, Germany, Italy
and along the Danube River. For more detailed information on the regions and project partners, we refer to the website here (https://directedproject.eu/about).

The **Risk-Tandem framework** is a novel approach combining risk management approaches and tools with iterative co-production methods and processes, in efforts to promote the co-design of fit-for-purpose solutions contributing toward strengthened risk governance alongside DRM and CCA stakeholders (Parviainen et al., 2025). The Risk-Tandem framework centres
around knowledge co-production for enhancing risk governance by enabling new and improved ways of facilitating transdisciplinary engagement, knowledge co-production, science-based dialogue and risk governance in the field of extreme climate and associated events. This integrated DRM and CCA risk-governance framework puts engagement and knowledge co-production, including capacity development, at its core by building on the Stockholm Environmental Institute's TANDEM framework (Daniels et al., 2020) and its applications (Bharwani et al., 2023). The Tandem framework (Daniels et al., 2020) is a practical,
non-prescriptive guide to co-design climate services by integrating diverse stakeholder goals and values, enhancing trust, and addressing multiple preferences, goals, capacities, and power dynamics. The framework has been applied in diverse settings with a range of users from municipalities in southern Africa, farmers in Indonesia, city planners in Sweden, and communities and institutions in a Colombian river basin. Each case demonstrated the framework's effectiveness in moving from 'useful' to 'usable' information, increasing institutional embedding, improving climate information uptake, and building capacity and
confidence among users and providers of climate information (Bharwani et al., 2024).

The Risk-Tandem framework bridges risk governance, communication, data and modelling by drawing on the International Risk Governance Council's Risk Governance Framework (Florin and Bürkler, 2018), the IIASA risk layering approach (Hochrainer-Stigler and Reiter, 2021), and the SHIELD model developed in the EPREssO project (Lauta et al., 2018).

Within a risk-layer approach, frequencies and corresponding losses of disaster events are grouped into risk-layers (e.g., low,
middle, high) and further related to generic risk instruments (e.g. risk reduction, risk financing and assistance). Losses in this context can be tangible or intangible, they can be measured in monetary terms based on market methods or not (Hochrainer-Stigler et al., 2023). Either way, the approach relies on the principle that different types of decision makers, e.g. in households, businesses, or the public sector are experiencing different contexts, and each of them should therefore adopt the most ap-

propriate strategy given their probabilistic hazard exposure, the cost efficiency of the risk-mitigating solutions they can use, and their access to financing instruments. Hence, through risk layering, gaps in individual risk measures as well as most appropriate instruments to increase resilience can be identified, both from a quantitative as well as a governance perspective (Hochrainer-Stigler et al., 2024).

The **Data Fabric** is proposed as a novel open-source federated data infrastructure, which enables stakeholders to consolidate and connect relevant data sources, models and information products across DRM and CCA application domains and institutional operating systems and languages. The software underlying the federated data infrastructure will be implemented based on open source solutions and extensions to it will be made available under an open source license. This allows an easy uptake to reuse or further develop the solutions in DIRECTED beyond the project for anyone interested. The platform will be federated in a way, such that not all data sources and models have to be hosted and run in the same infrastructure, but can be connected across different institutions and deployments via Application Programming Interfaces (APIs). This way, we address the limitations arising from restrictions on data sharing which is not always freely and openly possible in the context of DRM. The open-source federated data infrastructure will feature some open data and information products, but not necessarily all data will be openly available The infrastructure enhances information accessibility and offers bespoke visualisation and communication to decision-makers or relevant stakeholders. The interoperable infrastructure aims to identify flexible solutions around access rights, data sharing, taxonomies, and languages promoting open access and FAIR principles. The Data Fabric will build on suitable standards based on their application experiences, current use and acceptance and adopt or develop new best practices where needed. The data infrastructure flexibly integrates data and models already in place. Further, it provides enhanced functionality for connecting new models to the Data Fabric using APIs as well as to distributed data storage and (cloud) computing. Another building block is visualisation, which aims to enable a visual element of information transfer.

The Risk-Tandem Framework and the Data Fabric both complement one another to foster interoperability for disaster resilience. Both outputs aim to overcome silos, remove barriers between systems and link agents active in different domains of DRM and CCA, **Fig. 2**.

Within the DIRECTED project, all Real-World Laboratories (RWLs) have identified stakeholder needs and developed use cases for the Data Fabric, selecting back-end components for model integration. Currently, they are co-designing the front-end aspects of their Data Fabric implementations, while also developing user role-specific training modules. In the Capital Region of Denmark the primary challenge lies in future planning efforts, particularly involving flood simulations, impact assessments, and uncertainty analysis for adaptation measures around Roskilde Fjord. Key stakeholders include multiple municipalities in the region. For Emilia-Romagna Region, focus areas include real-time modelling and forecasting of pluvial and coastal flooding, as well as wildfire propagation. The main stakeholders driving this work are the regional civil protection authorities, firefighting services, and volunteer organizations. These groups prioritize inter- and intra-organizational communication and coordination, with a specific focus on engaging tourists in the region. The Danube Region has two focus areas in Vienna (Austria) and the Hungarian Zala region. This RWL addresses urban pluvial flooding and the impacts of climate-related floods and droughts on agriculture and tourism. Stakeholders range from local first responders to international insurance companies. A key objective is the development of consistent, cross-national data sets and models for DRM and CCA . The Rhine-Erft

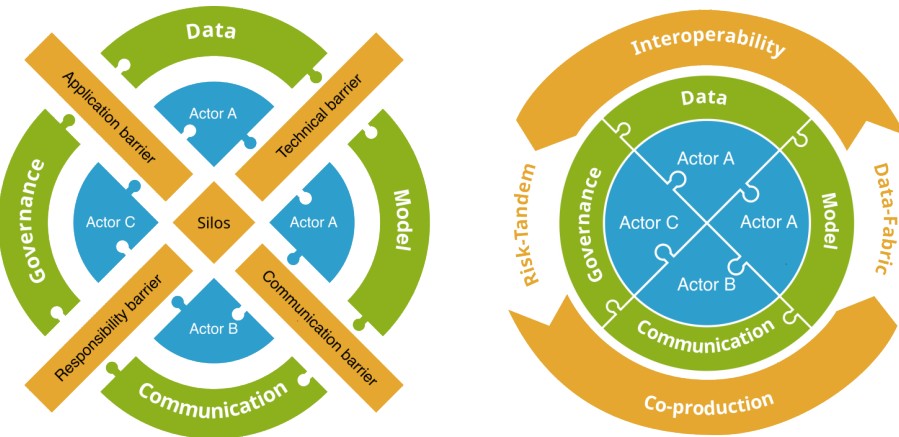

**Figure 2.** Risk-Tandem and Data Fabric are key outputs of a Real-Worl-Lab-based co-production process to foster interoperability between data, models, communication and governance by removing barriers and silos (left) and connecting actors in DRM and CCA (right).

Region focuses on improving communication and coordination among stakeholders, including the Waterboard Erftverband, municipalities, and emergency management services. Current efforts involve integrating social vulnerability into cost-benefit analyses for climate change adaptation, with a focus on strengthening governance at all levels. The RWL approach was specifically chosen for its transdisciplinary research mode which supports knowledge co-production and its focus on experimentation which facilitates testing and refinement of the Risk-Tandem and the Data Fabric (Schäpke et al., 2018). Another feature of this approach is the motivation to investigate real-world problems that are directly relevant to citizens and interest groups in those regions (Bergmann et al., 2021; Schäpke et al., 2018). Thus, the situatedness of the DIRECTED RWLs in specific geographical, ecological and social contexts provide the necessary focus for the inclusion of local knowledge and tailored engagement with interest groups. In addition, the DIRECTED RWLs are co-hosted by local authorities or other relevant actors in DRM and CCA risk management and disaster risk scholars. This joint responsibility supports building capacity of 'co-production facilitators' to ensure longevity and sustainability of activities beyond the duration of the DIRECTED project. The RWLs, led by practice-based hosts and supported by researchers/ scientists, act as a testing ground for co-developing and refining the Risk-Tandem Framework and the Data Fabric and benefit from strengthening stakeholder relationships through knowledge co-production leading to enhanced data and model, information and communication and governance interoperability. The transdisciplinary co-production process in the RWLs will serve to strengthen the capacity of Real World Lab hosts and enhances stakeholder and community engagement on DRM and CCA, to support the development of both outputs and to stimulate real-world partnerships and collaboration that will last beyond the project.

The Risk-Tandem framework is co-developed and tested in RWLs and refined based on the feedback and needs of RWLs. New insights generated from the process are expected to support RWL participants in their strategic (integrated) decision-making at long-term and immediate time scales and help break down silos between technical and political authorities at all levels, e.g. organisations, sectors and disciplines. Experiences from the co-production process in the RWLs emphasize that the

implementation of the Risk-Tandem framework is resource-intensive. Its implementation within a co-production approach requires continuous engagement of all stakeholders in discussions and continuous adjustments for balancing between a theoretical approach that requires practical implementation and risk governance practice. Instead of following a standard implementation approach, customizable and flexible application concepts are recommended to meet the case-specific stakeholder needs and shared challenges. Risk-Tandem facilitates a theory-informed approach for transdisciplinary co-production of risk reduction

and climate change adaptation strategies (Parviainen et al., 2025). In practice, this refers to the co-development of creative methodologies to co-explore contextual risk issues (including via the use of tabletop exercises, serious games, and visioning exercises), supporting the phases of co-design during which fit-for-purpose interventions are developed alongside RWL stakeholders. Interventions will prioritize issues of risk governance such as stakeholder engagement and coordination, and risk communication, for instance, improving public risk communication and -awareness. Local leadership by RWL hosts based in

different DRM and CCA agencies in combination with support from researchers/ scientists ensures that all Risk-Tandem activities and resulting solutions are rooted within local issues and priorities. To support this practical application of Risk-Tandem, RWL hosts are trained to build their collaborative, systems-thinking, creative and reflexive capacities and related facilitation, research and design skills to implement knowledge co-production in their RWLs (Cumiskey et al., 2025). For example, one host, the Civil Protection Agency of the Emilia-Romagna Region, benefited from training on systems-mapping exercises and

serious games. This increased their capacity to design and facilitate interactive workshop activities to map governance gaps, perspectives, needs and priorities for their RWL stakeholders. This then led to designing and implementing an intervention in their RWL For example, a flood simulation exercise with multiple stakeholders including volunteers to test the flood mapping tools being co-designed in the Data Fabric for extreme climate events and to better understand communication needs and flows and decision-making between agencies for different risk management actions, e.g. emergency response and urban

planning. The increased effort of this interdisciplinary co-production approach thus also leads to further development of the local actors and stakeholders and a deeper understanding of the factors and interrelationships that are crucial for the successful implementation of effective measures.

The stakeholder engagement and knowledge co-production processes will enable the co-production of the Data Fabric to meet specific needs in each RWL. As such, the Data Fabric will be developed particularly along the requirements and pain

points of the RWLs, asking questions, e.g. what information is needed to support robust decisions? Where are the gaps to be filled? What are the issues hindering the wide acceptance of truly interoperable data and models? Where do these intersect with barriers related to communication and governance? Additionally, the Risk-Tandem Framework in RWLs can identify governance barriers and enablers to ensure that the integrated data and modelling infrastructure can be embedded into institutional systems and result in improved risk analysis, communication and management decisions for DRM and CCA.

**4 Perspectives summary**

Implementing disaster risk management and climate change adaptation measures needs alignment but is complicated by barriers and gaps at different levels including governance structures and processes, communication and knowledge exchange among

actors as well as data and models. In this perspective paper we discuss challenges associated with interoperability between these components of integrated DRM and CCA.

To promote the application of data in planning or decision-making contexts - and to ensure it remains fit for purpose and context - multi-scalar knowledge exchange and communication between actors is essential. This includes inclusive dialogues and communication between DRM and CCA communities across multiple levels, such as resolving issues in understanding early warnings or seasonal forecasts and translating relevant information into effective and coordinated actions. In this regard, tailored and translated model-based information such as flood forecasts, disaster risk assessments, climate projections and

cost-benefit analyses can play a critical role in decision-support in different phases of the integrated DRM and CCA cycle.

    Governance interoperability aims to overcome siloed or fragmented knowledge- and information-sharing between actors across various political and societal levels, sectors, and disciplines, while leveraging synergies and multi-risk thinking to support integrated practices and policies. More specifically, governance interoperability facilitates data and information sharing across knowledge repositories supports a critical reflection of the societal implications of management options based on stake-

holder and public participation, and aims for more coherence of processes and responsibilities across governance levels and domains. As such, viewing risk governance through the lens of interoperability offers a means to address the governance challenges by linking them with information, communication, data and modelling challenges to find opportunities for collaboration, synergy and integration across DRM and CCA knowledge and governance systems.

    We argue that inclusive transdisciplinary knowledge co-production processes are the key to fostering interoperability be-

tween these systems to manage complex and interconnected climate and disaster risks. Knowledge co-production, however, is not a silver bullet. Its implementation remains challenging for numerous reasons (i) it is resource-intensive due to its iterative, reflexive, non-linear nature (Polk, 2015); (ii) it is transdisciplinary at its core, which involves a wide range of stakeholders across the science-society interface, representing a diversity of disciplines, sectors, skills and knowledge types (Norström et al., 2020); (iii) it can surface or compound inequalities (Turnhout et al., 2020); and, thus, it ideally requires actors with a particular

set of skills to co-design and carry out the process, including facilitators or knowledge brokers (Cvitanovic et al., 2016). Therefore, knowledge co-production processes should be understood as context-led solutions to complex challenges (Norström et al., 2020), that require continuous nurturing, learning and adapting to embed in multi-level/ collaborative/ inclusive risk governance systems. In addition, and given that knowledge co-production has been critiqued due to its lack of practical methodologies nor empirical evidence demonstrating impact (Miller and Wyborn, 2020), a structured and well-monitored approach is required.

The application of knowledge co-production processes with multiple DRM and CCA stakeholders can enhance knowledge exchange, learning, practice and policy, however, there is a limit to the change it can achieve. Such processes act as a space to develop shared goals, visions and ambitions, and conduct the groundwork to advocate necessary changes to regulatory regimes, organisational mandates or funding mechanisms for DRM and CCA. As such, the success of the proposed approach is dependent on other contextual factors which are beyond its control, such as the political landscape and conflict. However,

the combined focus on enabling governance, information and communication, and data and model interoperability aims to strengthen stakeholders' collective capacity to continuously adapt and respond to the changing context and associated needs/gaps for DRM and CCA. The Risk-Tandem framework and the Data Fabric are applicable in different contexts, sustained

by the overarching international policy context, such as the Sendai Framework for disaster risk reduction, the UN Sustainable Development Goals, and the Paris Agreement. The core premise of the underlying co-production approach is the need to be tailored to the application context. It is thus valid in various geographies and decision domains requiring transdisciplinary participation between a wide range of actors across the science-society interface. For example, the Tandem approach has been applied in both European contexts and beyond, in Africa, Asia, and Latin America. Through the DIRECTED project we intend to advance the required methods and tools and provide novel outputs for the DRM and CCA community to implement and fruitfully replicate the recommended co-creation process in RWL settings globally.

*Author contributions.* KS coordinated and led the writing and revision of the paper in close collaboration with PJS, BG, LC, SB, JP, CK, VWH, MD, TI and MS, All authors contributed to the conceptualisation of the paper, to the discussion on the content, text and ideas

*Competing interests.* At least one of the (co-)authors is a member of the editorial board of Natural Hazards and Earth System Sciences. The authors also have no other competing interests to declare.

*Disclaimer.* TEXT

*Acknowledgements.* DIRECTED receives funding from the European Union's Horizon Europe research and innovation programme (grant agreement no. 101073978). Associate partners SEI Oxford and Oasis Hub are funded by Innovate UK and ETH Zurich is funded by The State Secretariat for Education, Research and Innovations (SERI), Switzerland. Additional personal funding to add for any author?

**References**

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
