# Peer review of "Invited perspectives: Fostering interoperability of data, models, communication and governance for disaster resilience through transdisciplinary knowledge co-production"

_Natural Hazards and Earth System Sciences, 2024_

## Referee Comment (RC1)

**Review of the paper "Fostering interoperability of data, models, communication and governance for disaster resilience through transdisciplinary knowledge co-production"**

The paper discusses how disaster risk management (DRM) and climate change adaptation (CCA) are hindered by a lack of interoperability between data, models, communication, and governance. It provides a comprehensive overview of the technical, legal, operational, communicative and institutional barriers hindering effective responses. To overcome these barriers in the domains of data and models, communication, and governance, the authors suggest a transdisciplinary approach. They introduce frameworks such as a Risk-Tandem Framework or a Data Fabric to improve interoperability and facilitate knowledge co-production, aiming to enhance disaster resilience through integrated systems and governance.

The paper addresses the importance of improving disaster resilience by highlighting gaps in interoperability across various systems, including data, models, communication, and governance. This comprehensive, multi-dimensional approach provides value to a diverse range of stakeholders. However, the paper lacks depth regarding the practicalities of implementation. This is partly due to the fact that tests of the Risk Tandem and the Data Fabric are still pending. However, the transdisciplinary approach is not only costly in terms of resources, but also terms of the willingness of participants on all levels. In practice, securing the necessary resources and fostering cooperation between stakeholders could be challenging, even if highly desirable. The authors should elaborate in more detail, how participants and institutions can be motivated. In general, the paper would benefit from examples or practical instructions on how to overcome the barriers mentioned.

One aspect that could be discussed in addition is that different aspects are often relevant for different stakeholders. For example, a scientifically robust model is based on many parameters. However, only a fraction of these are relevant for political decision-makers. However, this does not mean that the model can be reduced to these few parameters. Nore that specific parameters are meaningful without context. Indeed, transdisciplinary product development is a meaningful tool to overcome such challenges.

In addition, I have listed some minor comments concerning different aspects throughout the paper.

Line 33

Please provide a source for the statement made in the sentence "In the absence of historic flood observation or due to a lack of local flood experiences [...]."

Line 199 to 203

The content described here seems repetitive.

Line 313

I doubt that running different models necessarily enhances trust. It can also lead to confusion due to conflicting results that do not always point to different realities but may instead be caused by model effects. To get to the bottom of such effects requires in-depth knowledge of the model. This does not seem to me to be the desirable goal, but rather that trust is built between the different stakeholders.

Line 324

Please specify what you envision by "creative and interdisciplinary approaches". In the current form, this is very ambiguous.

Line 353

Can you provide an example or specify the "specific roles and capacities needed"?

Line 381

There is something missing at the end of this line.

Line 435

You argue that the process is resource-intensive. In my opinion, this primarily comes from the transdisciplinary approach. Therewith, i) and ii) are at least linked and other reasons for the resource intensiveness should be added.

---

## Author Comment (AC1)

We thank the reviewer for the thorough evaluation of our manuscript and the helpful comments. We respond to these comments as follows:

Review of the paper "Fostering interoperability of data, models, communication and governance for disaster resilience through transdisciplinary knowledge co-production" The paper discusses how disaster risk management (DRM) and climate change adaptation (CCA) are hindered by a lack of interoperability between data, models, communication, and governance. It provides a comprehensive overview of the technical, legal, operational, communicative and institutional barriers hindering effective responses. To overcome these barriers in the domains of data and models, communication, and governance, the authors suggest a transdisciplinary approach. They introduce frameworks such as a Risk-Tandem Framework or a Data Fabric to improve interoperability and facilitate knowledge co-production, aiming to enhance disaster resilience through integrated systems and governance.

[1] The paper addresses the importance of improving disaster resilience by highlighting gaps in interoperability across various systems, including data, models, communication, and governance. This comprehensive, multi-dimensional approach provides value to a diverse range of stakeholders. However, the paper lacks depth regarding the practicalities of implementation. This is partly due to the fact that tests of the Risk Tandem and the Data Fabric are still pending. However, the transdisciplinary approach is not only costly in terms of resources, but also terms of the willingness of participants on all levels. In practice, securing the necessary resources and fostering cooperation between stakeholders could be challenging, even if highly desirable. The authors should elaborate in more detail, how participants and institutions can be motivated. In general, the paper would benefit from examples or practical instructions on how to overcome the barriers mentioned.

The motivation of this invited perspective paper is to present our perception of the critical role of interoperability at different levels in multi-hazard risk management and climate change adaptation and to discuss trans-disciplinary knowledge co-production as a means to foster interoperability in a multi-dimensional approach. The intention of the paper is to put this idea to an open debate in the scientific community for collecting suggestions and further thoughts. It is beyond the scope of this invited perspective to provide comprehensive tests of the suggested tools (i.e. Data Fabric and Risk-TANDEM), which is still the subject of ongoing research. Further papers discussing these developments in detail and the practicalities of implementation are currently under review in other journals. We agree that our presentation would benefit from additional examples to illustrate the issues addressed and the underlying ideas of the concept and tools. As this point is also raised in other comments in the reviews, we make suggestions for examples in the answers to these comments, including also examples for practicalities of implementation, see [R1-7, R2-16] and willingness of participants [R2-16].

[2] One aspect that could be discussed in addition is that different aspects are often relevant for different stakeholders. For example, a scientifically robust model is based on many parameters. However, only a fraction of these are relevant for political decision-makers. However, this does not mean that the model can be reduced to these few parameters. Note that specific parameters are meaningful without context. Indeed, trans-disciplinary product development is a meaningful tool to overcome such challenges.

We agree that the importance of different aspects varies for different stakeholders. We will pick up this point with regard to models in [section 2.1 ll 165]:

"Furthermore, even if all data is interoperable and well documented by metadata, it remains challenging to use it in practice as users can rarely be experts in all the fields required to adequately judge the fitness for purpose. The crux of data and model interoperability rests on finding the right balance between model complexity and the granularity of input and output, harmonized with **stakeholder specific preferences** as well as collective contextual understanding and discernment. An overabundance of detail in data and metadata can obscure vital insights, and an overly intricate and complex model at one stage may be suboptimal to the performance of subsequent applications. The objective is to **reflect on important aspects for different stakeholders** and to supply the appropriate amount and level of data and contextual knowledge sharing necessary to inform the next stages in modelling or decision-making processes. While several technical solutions such as the definition of data standards are prerequisite for this objective, direct exchange between experts of connected models and data is often the only way to achieve maximum interoperability."

With regards to communication and knowledge transfer in [section 2.2 ll186]:
"Translating data and knowledge into action is often missing due to a lack of understanding of user needs and the integration of different **stakeholders perspectives and aspects**, knowledge and disciplines. This can compound the lack of collaboration between data providers themselves as well as with users and other **stakeholders** and lead to a dismissal of values and norms that inform decision-making and therefore affect the uptake and application of data and information as they may not be fit for both purpose and context."

In addition, interoperability is not only important in regard to models and communication and knowledge transfer but also in regard to governance processes that need, as we argue, to be jointly embedded within a knowledge co-production process to identify and overcome potential interoperability gaps. The identification of such gaps are a cornerstone for determining appropriate tools and processes to overcome them, either within models or governance processes or importantly between them. Therefore, we suggest to also add the aspect of stakeholder perspectives with regards to governance in [section 2.3 ll 244]:

"Therefore, stakeholder and public engagement in risk governance are motivated by the realisation that these groups provide crucial information for assessment from diverse standpoints and **perspectives**, including scientific knowledge and other knowledge systems (Fischhoff, 1995)."

In addition, I have listed some minor comments concerning different aspects throughout the paper.

[3] Line 33: Please provide a source for the statement made in the sentence "In the absence of historic flood observation or due to a lack of local flood experiences [...]."

We propose to add Kreibich, H., et al. 2017 (https://doi.org/10.5194/nhess-17-2075-2017, 2017) and Bertola et al. 2023 (https://doi.org/10.1038/s41561-023-01300-5) as a references for this statement.

[4] Line 199 to 203: The content described here seems repetitive.

We assume that this refers to repeating statements from the previous section 2.1 on data and model interoperability. We propose rephrasing ll 190 as follows:

"As such, technical data interoperability must be accompanied by efforts to support the interoperability of the information and communication channels and approaches seeking

to address these issues. Issues of communication also emerge from **data and model interoperability (cf. Challenge 1 in section 2.1)**

[5] Line 313: I doubt that running different models necessarily enhances trust. It can also lead to confusion due to conflicting results that do not always point to different realities but may instead be caused by model effects. To get to the bottom of such effects requires in-depth knowledge of the model. This does not seem to me to be the desirable goal, but rather that trust is built between the different stakeholders.

We realise that our statement is not clear. Building trust in the context of co-production is a multifaceted idea, which requires additional explanation. We think that in the context of our perspective paper two main aspects need to be considered: a) trust in models in the sense that the models are fit for the intended purpose, and b) building trust among stakeholders and decision makers in data and models who build their decision on the outcomes of available data and models.

Regarding a), we agree that running different models does not inherently enhance trust, in particular when outputs are highly uncertain and variable. At the same time, one of the main ways for scientist to build trust in models that are only hardly verifiable (such as risk models of the future, e.g. Merz et al. 2024) is to have several, distinct modelling approaches building up to shared evidence and a common understanding. For instance climate models may lead to very large variance in outputs for global temperatures for the same emission trajectories. Confidence in the climate models is established by pooling the information from several, distinct models. This confidence was build thanks to a large-scale interoperability effort lead by the CMIP team. We argue that this type of multi-model angle is not only required on the physical modelling side where it is already common, but through the whole chain of information. This also relates to the question what models can be used for which purpose. Indeed each model has to clearly state its limitations as well as advantages and additional models usually targeting a different set of questions to overcome some limitations of other models. In case of conflicting results one can then use the above approach of model ensembles or other approaches including climate storylines. We also argue that this complexity, while it requires careful communication, should not be hidden from different stakeholders. In fact, transparently communicating the uncertainty in the modelling by showing the outputs of different models not only increases the robustness of the process, but also builds trust.
However, regarding b) the communication part remains absolutely central to this and requires a lot of care. The proposed knowledge co-production process, i.e. connecting modellers, data providers and its end users, promotes discussions regarding different modelling approaches, and explores user needs vis-à-vis available information in a non-hierarchical manner (see Daniels, et al., 2020). This is expected to increase the usability and accessibility of information by clarifying potential errors, uncertainties and underpinning assumptions embedded in each model for users, in an effort to align available information with needs of planners and decision-makers. In other words, the process is an on-going negotiation between needs and what models can provide, which, insofar as uncertainty is accounted for, also generates trust in data through continuous and transdisciplinary engagement with it (Daniels, et al., 2020). Effective communication plays a pivotal role in establishing trust in DRM and CCA decisions. Uncertainties embedded in underlying modelling can be large, and uncertainty information are often difficult for stakeholders to understand. Since disclosure of uncertainties does not always increase trust and credibility in risk analyses (Doyle et al., 2019), information about uncertainty should be embedded in the co-production process and tailored to the specific audience and consider their perspectives, technical knowledge and concerns (Merz et al. 2024). For the paper we suggest to rephrase the text:

"A possible way to embed this in existing workflows is to build standardized and simplified data and model pipelines, which encapsulate some of the complexity and allow for the easy running of separate models addressing the same question. This allows comparing model outputs of different approaches, which can support building trust in the models to be fit for purpose. Importantly, the co-production process connects modellers, data providers and end users, promotes discussions regarding different modelling approaches, and explores user needs in contrast to available information (Daniels et al., 2020). In this process, effective communication is essential in establishing trust in DRM and CCA decisions. Uncertainties embedded in underlying data and models can be considerable. For stakeholders, uncertainties are often difficult to understand. Therefore, information about uncertainty should be embedded in the co-production process and tailored to the specific audience and consider their perspectives, e.g. Merz et al. 2024 for the example of flood hazard and risk modelling."

[6] Line 324: Please specify what you envision by "creative and interdisciplinary approaches". In the current form, this is very ambiguous.

We suggest to add the following sentences which pick-up examples from the papers cited for clarification:

"Interactivity is one of the key principles of knowledge co-production (Norström et al., 2020). Creative approaches can include interactive games to co-explore issues, language and terminology (Daniels et al. 2020) and participatory arts-based methods to support three-way communication between decision-makers, scientists and communities (Stewart, 2024). These aim to clarify and unpack complex and sometimes field-specific knowledges so that they become easily accessible for all stakeholder groups –for instance, with the help of interactive and game-based methods such as tabletop exercises, serious games, or "future-casting" and visioning."

[7]Line 353: Can you provide an example or specify the "specific roles and capacities needed"?

We propose to add the following example for specific  roles and capacities needed:

"These roles support trans-disciplinary knowledge co-production through their capacity to network, communicate, facilitate and convene multiple diverse stakeholders, navigate their different goals/values and synthesise and integrate/broker knowledge to influence policy or practice. This entails identifying champions and facilitators capable and interested in bridging connections between disciplines and knowledges involved (Bharwani, et al., 2024), in consideration of the skills and capacities required for them to enable co-production as described above."

[8] Line 381: There is something missing at the end of this line.

This is simply a punctuation error. The sentence should read: "Another building block is visualisation, which aims to enable a visual element of information transfer."

[9] Line 435: You argue that the process is resource-intensive. In my opinion, this primarily comes from the transdisciplinary approach. Therewith, i) and ii) are at least linked and other reasons for the resource intensiveness should be added.

Indeed these two aspects are linked, however, in our context we argue that the resource-intensiveness comes from (i) knowledge co-production that also happens within scientific disciplines (e.g. economic modelling of indirect risks due to hazard events through General Equilibrium Models and Agent -Based Modelling approaches to include long-term and short term-behaviour, respectively, see for example Botzen et al. (2019) or a specific hazard focus, e.g. flood events across different spatial scales and integration of results for a comprehensive flood risk analysis). So in other words, the iterative, reflexive, non-linear nature within a co-production process is already resource intensive even if it is not (yet) trans-disciplinary. In case, as we argue that it should be, that it is trans-disciplinary, additional resources are needed (that may not be needed for point (I)) but are referred to in ii) as for instance, to increase understanding between different scientific disciplines as well as stakeholders and the organization and orchestration of such tasks to manage the additional complexity for knowledge integration.

References:

Bharwani, S., Gerger Swartling, Å., André, K., Santos Santos, T. F., Salamanca, A., Biskupska, N., Takama, T., Järnberg, L., and Liu, A.: Co-designing in Tandem: Case study journeys to inspire and guide climate services, Climate Services, 35, 100503, https://doi.org/10.1016/j.cliser.2024.100503, 2024.

Botzen, W. W., Deschenes, O., & Sanders, M. (2019). The economic impacts of natural disasters: A review of models and empirical studies. Review of Environmental Economics and Policy. https://www.journals.uchicago.edu/doi/full/10.1093/reep/rez004

Doyle, E. E. H., Johnston, D. M., Smith, R., and Paton, D.: Communicating model uncertainty for natural hazards: A qualitative systematic thematic review, International Journal of Disaster Risk Reduction, 33, 449–476, https://doi.org/10.1016/j.ijdrr.2018.10.023, 2019.

Merz, B., Blöschl, G., Jüpner, R., Kreibich, H., Schröter, K., and Vorogushyn, S.: Invited perspectives: safeguarding the usability and credibility of flood hazard and risk assessments, Natural Hazards and Earth System Sciences, 24, 4015–4030, https://doi.org/10.5194/nhess-24-4015-2024, 2024.

---

## Author Comment (AC2)

We thank the reviewer for the thorough evaluation of our manuscript and the helpful comments. We respond to these comments as follows:

This paper gives a broad consideration of how to improve interoperability of data, models, communication and governance for disaster risk management and climate adaptation. The paper starts with a comprehensive and thoughtful review of well-established ideas. I found it instructive to think about interoperability with regard to communication and governance, which I hadn't considered before.

[1] My main reflections are that more explanation of key terms or projects is needed because without this the narrative is quite difficult to follow. Real examples would also be extremely helpful throughout. A more detailed description of the DIRECTED project is warranted because at the moment it appears to be a fairly high level overview of what is planned rather than an active project tackling the issues that have been identified earlier in the paper. Detailed comments follow below:

We agree that our presentation would benefit from additional explanation of key terms and examples to illustrate the issues addressed and the underlying ideas of the proposed concepts. This point is also mentioned in other comments below and has also been raised by reviewer #1. Therefore, we refer to our answers to these points where we make suggestions for possible examples [R1-6, R1-7, R2-10, R2-16]

The paper does not intend to describe the DIRECTED project in depth but we will add further details as described in our answer to comment [R2, 17]. We will also elaborate on the description of the innovations for interoperability, Data-Fabric and Risk-Tandem in section 3.2

[2] Title: In the title or early on in the paper it would be helpful to indicate that the focus of the paper (if I understand correctly) is Europe. This would help to set what follows in context.

We do not have a particular focus on Europe per se. However, due to the location of the real-world labs in Europe, the conditions there are at the centre of our considerations, particularly at the governance level. The transferability of results for the real-world labs will be bounded to western cultural settings. The methods should be applicable in other contexts as well, given the policy context of e.g. the SENDAI Framework. The core premise of the co-production approach is that it is tailored to context and thus applicable in a range of geographies and decision domains requiring trans-discplinary participation between a wide range actors across the science-society interface. As such, e.g. the Tandem approach has been applied in both European contexts and beyond, in Africa, Asia and Latin America. Also with regard to models, data and knowledge transfer, the approaches we discuss are quite universal and can also be transferred to other regions.

[3] Line 3: Consider whether 'emerging' is the appropriate word here. I would argue that many of the complications that are described in the paper are widely known about.

We suggest to replace the word 'emerging' with 'manifest'.

[4] Line 4: I would argue there are other factors like a lack of investment and capacity in local scientific organisations, which may have implications for interoperability (in some countries). This connects to the need to be clear about the examples that have informed the development of the ideas on which the paper is based.

Yes, a lack of investment and capacity in local organisations may result in interoperability issues, but we think based on the framing of the three overarching challenges these issues are also covered. Through the work with stakeholders in our real-world labs and in exchange with other research initiatives in the Horizon Europe program as well as The Mission on Adaptation to Climate Change we have learned that organisations often lack long term investment in dedicated DRM and CCA capacity building. Even institutions that are actively involved in research and innovation may not have organisation spanning DRM and CCA capacities. We observe that special project related roles within these organisations are created, but are often not permanent. Access to knowledge is kept in "Silos" and does not permeate the organisations. The DIRECTED co-production approach acknowledges this reality and builds bridges and lasting relationships between actors from different disciplines within organisations and across boarders. Resources such as the Data Fabric with integrated models and the Risk-Tandem Framework with guiding documents and training modules will be made available long term.

[5] Line 19: comma missing between 'drought' and 'heavy rainfall'

We will add the comma.

[6] Line 37-38: A little more information regarding what 'limited imagination' pertains to would be helpful here. I would argue that decision makers not understanding the information or information not being action-focused can also be barriers.

Good point. We will prick-up this suggestions and propose to remove reference to 'limited imagination' and instead refer to the fact that 'whilst learning from other regions facing similar risks is possible, a lack of 'lived experience' and the uncertainty of the true impact of the many interacting complex factors at play in different locales, could still prove challenging to act upon.

[7] Line 59: Would be helpful to give examples of the kinds of hazard variables that might be affected here, especially when taking a multi-hazard view (e.g. the impact of extreme precipitation on slope stability).

We agree and suggest to add examples as follows

"Therefore, information on local climate change impacts on hazard variables **such as rainfall intensity and volume, prolonged dry spells or higher extreme temperatures** needs to be produced, but importantly also needs to be accessible and embedded in practical planning processes in a structured and transparent way.

[8] Line 60: I wonder if there's another word that could be used instead of 'adversely' here just because it's not often used at the start of a sentence. 'Responsibilities… exchange of information and communication, *which has an adverse effect*' might work better if that's what your intended meaning is.

We agree and suggest to delete the word 'Adversely'.

[9] Line 62: 'homogeneous' – explain what the homogeneity relates to

In this context 'homogeneous' relates to how risk communication approaches by authorities typically use one message to communicate to multiple audiences e.g. early warning alerts, rather

than a tailored or targeted approach to meet the needs of different audiences. See also examples in [R2-10]

[10] Line 64: General point – some examples throughout this section would help to reinforce and/or illustrate the points you are making. It would also help to root the discussion in the geographic context of the paper (or highlight contrasting situations in other contexts if appropriate).

We propose to add some examples for better illustration:

 "Adversely, Responsibilities for planning, implementation and management are distributed across administrative offices which complicates and impedes the exchange of information and communication. In addition, the communication of climate and disaster risks to the public is typically one-way and homogeneous but can be enhanced through two-way dialogues that identify, engage and consult with specific stakeholders to develop tailored communications (requiring detailed analysis of the composition of different actors within an audience) and three-way participation where communication becomes a collective and continuous process of knowledge production between citizens, science and decision-makers (Stewart, 2024) e.g. using methods such as art- and citizen science, interactive games, role plays etc. Key differences and examples of the continuum between one-way, two-way, and three-way communications are provided by Stewart (2024). Moving from product to process, Tandem (Daniels et al., 2020) was applied in a southern African urban context addressing adaptation and disaster risk challenges in peri-urban areas using a trans-disciplinary 'Learning Lab' approach and 'embedded researchers' to bring stakeholders together to identify and prioritize challenges and co-create solutions and creating long lasting relationships, which support ongoing networks such as the public-private multi-stakeholder partnership, the Lusaka Water Security Initiative (LuWSi). Recent applications (Bharwani et al., 2024) diversified Tandem's use in different socioeconomic settings and decision domains. A rural Indonesian community of smallholder coffee and cacao farmers co-created weather forecasts with the national meteorological office to tailor farmer field school curricula with local ecological knowledge, concepts and terminology. In Sweden urban planners, meteorological scenario modellers, hydro-climatologists and city officials co-explored compound events related to flooding (cloudburst events and spring floods) as well as heatwave scenarios to inform the city's 2024 Stockholm's Environmental Programme (2020-2023) and the Climate Adaptation Action Plan (2022-2025). In Colombia, a participatory group, the river basin council, including representatives from local and regional communities and institutions addressing water scarcity and inequitable access (farmers, municipalities, NGOs, indigenous populations and the private sector), co-designed a graphical web tool interface that translated hydro-meteorological data into accessible, relevant and usable information and language for basin planning, that continues to be used today. All of these processes enhanced information interoperability, as well as the capacity and confidence of stakeholders to work with and recognize the limits of climate information (Bharwani *et al.,* 2024)."

[11] Line 75: Here, it would be worth considering what the implications of these consistencies would be on decision makers (e.g. confusion and decision making that isn't joined up across borders?)

We agree and suggest to follow the suggestion by amending the text:

"One example is the production of flood hazard and risk maps during the implementation of the European Floods Directive (2007/60/EC, 2007) with diverging definitions regarding the extreme flood scenario, which leads to inconsistencies in hazard and risk information across federal state or national borders and eventually causes confusion in decision making for trans-boundary flood risk management."

[12] Line 84: Give full name for INSPIRE

We will spell out INSPIRE as: Infrastructure for Spatial Information in the European Community (INSPIRE)

[13] Line134-135: Is there an example that could be given here (if it's possible to do that)?

We think it is rather the norm than an exception that licences restrict the sharing of data. If need we propose to add the following examples:

"Licensing terms can restrict the use and sharing of data, since for instance scientific users might not have a legal team to handle subtleties of non-open licenses (e.g., CC BY-NC-SA), non-commercial users might not have the financial means to pay for expensive data licenses, or commercial users might not want their derived products to have open licenses (e.g., CC BY-SA). Furthermore, privacy regulations, such as those mandated by the General Data Protection Regulation (Regulation (EU) 2016/679), while necessary to protect basic human rights, impose additional layers of complexity, often requiring the anonymisation of data. For instance, simply revealing the total number of minors in a vulnerable region, while potentially valuable for risk assessment models, could also be exploited by Human trafficants, and thus requires extra care and consideration before sharing."

[14] Section 2.2: Some illustrative examples would help to root this section in the European context.

We have added some examples in response to comment [R2-10]

[15] Line 247: Explain what ESPREssO is to help readers who may not be familiar with the project.

We suggest to add the following short explanation for the ESPREssO project:

Enhancing synergies for disaster prevention in the European Union (ESPREssO) project addressed the integration of DRR and CCA.

[16] Section 3.1: Again, some real examples would bring this section to life. At the moment it feels quite 'hypothetical'. It would be great to include some vignettes of where these sorts of things have been attempted and it has worked/not worked.

In response to the other reviewer 's comments we propose to add several examples and explanation, wee [R1-5, 6, 7]. Regarding Data and Models we included a standardized resilience measurement tool that tries to overcome these interoperability gaps. We suggest to included the following text (ll 308):

"For example, the Flood Resilience Measurement for Communities (FRMC) tool not only quantitatively assesses different sources of resilience against flood risks on the community level but also highlight strengths and weaknesses in community resilience using different perspectives that

can be visualized, arranged and displayed flexibly according to the user needs (Zurich Flood Resilience Alliance 2019; https://zcralliance.org/resources/item/the-flood-resilience-measurement-for-communities-frmc/).”

[17] Line 363: Some introductory information about DIRECTED would be very helpful here (e.g. when it started, objectives, geographic focus) because it is only mentioned briefly at the start of the paper and there's a lot of content before it is presented here. Is it being used in practice? Who is involved?

Our proposition is to add the following explanation of the DIRECTED Project

“The DIRECTED project with its trans-disciplinary consortium of researchers, practitioners and industry partners has begun its work in late 2022 and is scheduled to conclude in September 2026. The project's main objective is strengthen the disaster resilience of societies by overcoming existing silos in DRM and CCA.  DIRECTED aims to strengthen interoperability between modelling tools used in DRM and CCA to improve communication and governance processes. These innovations for interoperability are co-developed and implemented in an iterative approach with local project partners and stakeholder groups in four European Real Word Labs in Denmark, Germany, Italy and along the Danube River. For more detailed information on the regions and project partners, we refer to the website here (https://directedproject.eu/#about).”

[18] Line 365: explain what is meant by 'new and improved'

Here 'new and improved ways' refers to the processes, methods and tools being applied and tested to co-design technical and governance DRM and CCA solutions (in the Real World Labs) within the Risk-Tandem framework. However, for clarity we propose to edit the sentence to the following and have added a reference to a paper (in publication) that can provide more details on the methods and tools.

“The Risk-Tandem framework is a novel approach combining risk management approaches and tools with iterative co-production methods and processes, in efforts to promote the co-design of fit-for-purpose solutions contributing toward strengthened risk governance alongside DRM and CCA stakeholders (see Parviainen et al. 2024, in review).”

[19] Line 369: a short explainer of the TANDEM framework would be helpful here. Also the IIASA risk layering approach and the SHIELD model should be explained.

We suggest to add the following explanations for these frameworks and approaches:

“The Tandem framework (Daniels et al., 2020) is a practical, non-prescriptive guide to co-design climate services by integrating diverse stakeholder goals and values, enhancing trust, and addressing multiple preferences, goals, capacities, and power dynamics. The framework has been applied in diverse settings with a range of users from municipalities in southern Africa, farmers in Indonesia, city planners in Sweden, and communities and institutions in a Colombian river basin. Each case demonstrated the framework's effectiveness in moving from 'useful' to 'usable' information, increasing institutional embedding, improving climate information uptake, and building capacity and confidence among users and providers of climate information (Bharwani et al., 2024).

Within a risk-layer approach, frequencies and corresponding losses of disaster events are grouped into risk-layers (e.g., low, middle, high) and further related to generic risk instruments (e.g. risk reduction, risk financing and assistance). Losses in this context can be tangible or intangible, they can be measured in monetary terms based on market methods or not (Hochrainer-Stigler et al. 2023). Either way, the approach relies on the principle that different types of decision makers—e.g., in households, businesses, or the public sector—are experiencing different contexts, and each of them should therefore adopt the most appropriate strategy given their probabilistic hazard exposure, the cost efficiency of the risk-mitigating solutions they can use, and their access to financing instruments. Hence, through risk layering, gaps in individual risk measures as well as most appropriate instruments to increase resilience can be identified, both from a quantitative as well as a governance perspective (Hochrainer-Stigler et al. 2024)."

[20] Line 373:  More explanation of what an 'open-source federated data infrastructure' is would be useful here. What does 'federated' mean in this context? In my experience, making information open-source can sometimes be problematic for individuals and/or organisations. It might be worth considering the challenges this might present to what you are proposing.

We suggest to add the following additional explanation to the text:

"The software underlying the federated data infrastructure will be implemented based on open source solutions and extensions to it will be made available under an open source license. This allows an easy uptake to reuse or further develop the solutions in DIRECTED beyond the project for anyone interested. The platform will be federated in a way, such that not all data sources and models have to be hosted and run in the same infrastructure, but can be connected across different institutions and deployments via Application Programming Interfaces  (APIs). This way, we address the limitations arising from restrictions on data sharing which is not always freely and openly possible in the context of DRM. The open source federated data infrastructure will feature some open data and information products, but not necessarily all data will be openly available."

[21] Line 380: expand APIs

We will expand APIs: Application Programming Interfaces (APIs)

[22] Line 385: more information on the individual real world laboratories is necessary here, e.g. the main challenges the approaches would help to overcome in each setting, who's involved, progress so far, etc. This would help to connect this section to the earlier sections.

We suggest to add the following information about the real world laboratories, but are open to shorten, of this is too extensive.

"All Real-World Laboratories (RWLs) have identified stakeholder needs and developed use cases for the Data Fabric, selecting back-end components for model integration. Currently, they are co-designing the front-end aspects of their Data Fabric implementations, while also developing user role-specific training modules. In the Capital Region of Denmark the primary challenge lies in future planning efforts, particularly involving flood simulations, impact assessments, and uncertainty analysis for adaptation measures around Roskilde Fjord. Key stakeholders include multiple municipalities in the region. For Emilia-Romagna Region, focus areas include real-time modelling and forecasting of pluvial and coastal flooding, as well as wildfire propagation. The main

stakeholders driving this work are the regional civil protection authorities, firefighting services, and volunteer organizations. These groups prioritize inter- and intra-organizational communication and coordination, with a specific focus on engaging tourists in the region. The Danube Region has two focus areas in Vienna (Austria) and the Hungarian Zala region. This RWL addresses urban pluvial flooding and the impacts of climate-related floods and droughts on agriculture and tourism. Stakeholders range from local first responders to international insurance companies. A key objective is the development of consistent, cross-national data sets and models for disaster risk management (DRM) and climate change adaptation (CCA). The Rhine-Erft Region focuses on improving communication and coordination among stakeholders, including the Waterboard Erftverband, municipalities, and emergency management services. Current efforts involve integrating social vulnerability into cost-benefit analyses for climate change adaptation, with a focus on strengthening governance at all levels."

References

Hochrainer-Stigler, S., Trogrlić, R. Š., Reiter, K., Ward, P. J., de Ruiter, M. C., Duncan, M. J., ... & Gottardo, S. (2023). Toward a framework for systemic multi-hazard and multi-risk assessment and management. IScience, 26(5).
https://www.sciencedirect.com/science/article/pii/S2589004223008131

Hochrainer-Stigler, S., Deubelli-Hwang, T. M., Parviainen, J., Cumiskey, L., Schweizer, P. J., & Dieckmann, U. (2024). Managing systemic risk through transformative change: Combining systemic risk analysis with knowledge co-production. One Earth, 7(5), 771-781.
https://www.cell.com/one-earth/fulltext/S2590-3322(24)00204-5

Parviainen, J., Hochrainer-Stigler, S., Cumiskey, L., Bharwani, S., Schweizer, P.-J., Hofbauer, B., Cubie,D., 2024. Risk-Tandem: an iterative framework for combining risk governance and knowledge co-production toward integrated disaster risk management and climate change adaptation. In Review. International Journal of Disaster Risk Reduction

---

## Author Response (AR2)

We thank the reviewer and the editor for the thorough evaluation of our revised manuscript and the helpful comments. We respond to these comments as follows:

Reviewer #3:

The manuscript explores how enhancing interoperability across data, models, communication, and governance through knowledge co-production can strengthen disaster risk management and climate change adaptation, but it requires clearer contextual framing to reflect its predominantly European practical perspective and limitations in broader applicability. Apart from that, I'll focus my assessment on evaluating the authors' responses to the existing reviewers' comments.
The authors have made considerable efforts to respond to the reviewers' comments using a constructive approach. In their responses, they show a good understanding of the points raised by the reviewers.

Thank you for the overall positive assessment of our implemented changes. Even though we do not have a particular focus on Europe per se, as clarified in our response to the previous review #2, we recognize that this should be better explained in the paper. We would like to stress, that in the previous revision we have added application examples of underlying concepts and methods (e.g Tandem and Real-World-Lab approaches) from different geographies globally (ll 69 to 86), demonstrating that the proposed approach is not limited to Europe.
In addition we suggest to further expand on this in the introduction and perspective summary as follows:

Introduction, ll 128 (suggested additions to the text in bold):

"In this perspective paper, we discuss interoperability challenges for DRM and CCA by taking a detailed look at data and models, information and communication, and governance systems (Chapter 2). On this basis, we propose recommendations for overcoming these challenges (Chapter 3) based on research and development work carried out in the inter- and transdisciplinary EU innovation project DIRECTED which aims to reduce vulnerability to extreme weather events and foster disaster-resilient societies by promoting interoperability between DRM and CCA. **While the learning from real-world-labs within the DIRECTED project is based on the specific conditions given in these European settings, the methods proposed will be applicable in other geographical and cultural contexts.** We summarise our perspectives on interoperability for disaster resilience through transdisciplinary knowledge co-production (Chapter 4)."

Perspective summary, ll 541 (suggested additions to the text in bold)

"However, the combined focus on enabling governance, information and communication, and data and model interoperability aims to strengthen stakeholders' collective capacity to continuously adapt and respond to the changing context and associated needs/-gaps for DRM and CCA. **The Risk-Tandem framework and the Data Fabric are applicable in different contexts, sustained by the overarching international policy context, such as the Sendai Framework for disaster risk reduction, the UN Sustainable Development Goals, and the Paris Agreement. The core premise of the underlying co-production approach is the need to be tailored to the application context. It is thus valid in various geographies and decision domains requiring transdisciplinary participation between a wide range of actors across the science-society interface. For example, the Tandem approach has been applied in both European contexts and beyond, in Africa, Asia, and Latin America.** Through the DIRECTED project we intend to advance the required methods and tools and provide novel outputs for the DRM and CCA community to implement and fruitfully replicate the recommended co-creation process in RWL settings **globally.**

[1] That said, a few important points remain only partially addressed, in my opinion. For example, reviewer #1 raised concerns about the lack of depth regarding implementation practicalities and stakeholder motivation in transdisciplinary approaches (point [1]). While the authors acknowledge that this manuscript is not meant to provide comprehensive implementation details and refer to other publications under review, which is a vague argument in terms of scientific evidence, the rebuttal leans too heavily on deferral. This, in my opinion, leaves a noticeable gap in the manuscript, especially given the emphasis of this perspective piece on real-world application.

We understand the concern of the reviewer regarding practicalities of implementation and stakeholder motivation, which is indeed a critical issue for applying transdisciplinary approaches. We propose to elaborate on our experience in implementing the Risk-Tandem framework within the real world labs. We will draw on new insights gained within the recently published studies of Parviainen et al (2025) and Cumiskey et al (2025) which details the practicalities of capacity development for locally-led knowledge co-production in RWLs within the framework of Risk-Tandem
We suggest to include this additional contents in section 3.2 Innovations for interoperability (ll 496, suggested additions to the text in bold):

"The Risk-Tandem framework **is** co-developed and tested in RWLs and refined based on the feedback and needs of RWLs. New insights generated from the process are expected to support RWL participants in their strategic (integrated) decision-making at long-term and immediate time scales and help break down silos between technical and political authorities at all levels, e.g. organisations, sectors and disciplines. **Experiences from the co-production process in the RWLs emphasize that the implementation of the Risk-Tandem framework is resource-intensive. Its implementation within a co-production approach requires continuous engagement of all stakeholders in discussions and continuous adjustments for balancing between a theoretical approach that requires practical implementation and risk governance practice. Instead of following a standard implementation approach, customizable and flexible application concepts are recommended to meet the case-specific stakeholder needs and shared challenges. Risk-Tandem facilitates a theory-informed approach for transdisciplinary co-production of risk reduction and climate change adaptation strategies (Parviainen et al. 2025). In practice, this refers to the co-development of creative methodologies to co-explore contextual risk issues (including via the use of tabletop exercises, serious games, and visioning exercises), supporting the phases of co-design during which fit-for-purpose interventions are developed alongside RWL stakeholders. Interventions will prioritize issues of risk governance such as stakeholder engagement and coordination, and risk communication, for instance, improving public risk communication and -awareness.
Local leadership by RWL hosts based in different DRM and CCA agencies in combination with support from researchers/ scientists ensures that all Risk-Tandem activities and resulting solutions are rooted within local issues and priorities. To support this practical application of Risk-Tandem, RWL hosts are trained to build their collaborative, systems-thinking, creative and reflexive capacities and related facilitation, research and design skills to implement knowledge co-production in their RWLs (Cumiskey et al. 2025). For example, one host, the Civil Protection Agency of the Emilia-Romagna Region, benefited from training on systems-mapping exercises and serious games. This increased their capacity to design and facilitate interactive workshop activities to map governance gaps, perspectives, needs and priorities for their RWL stakeholders. This then led to designing and implementing an intervention in their RWL For example, a flood simulation exercise with multiple stakeholders including volunteers to test the flood mapping tools being co-designed in the Data Fabric for extreme climate events and to better understand communication needs and flows and**

**decision-making between agencies for different risk management actions, e.g. emergency response and urban planning. The increased effort of this interdisciplinary co-production approach thus also leads to further development of the local actors and stakeholders and a deeper understanding of the factors and interrelationships that are crucial for the successful implementation of effective measures."**

[2] Another example is point [2] of reviewer #1, where the authors reacted by tweaking a few words in the text to address something that clearly demanded more depth. A more effective response and corresponding change in the manuscript would be to summarise actionable strategies briefly or include a short "snapshot" or box within the manuscript offering practical insights or motivational mechanisms drawn from experience. While the authors comment on balancing model complexity and stakeholder needs, the point would be more compelling if supported by a specific example, such as how different types of flood models may serve the needs of emergency responders versus urban planners (only an example).

We address this point in response to the previous comment [1] where we describe the practical details of the Risk-Tandem-Framework implementation. In the suggested additions to the text we describe how various stakeholder perspectives and preferences are taken into account during the implementation process in the RWL and we also provide an example for the co-design of the data fabric to provide tailored information which is useful for specific risk management actions.

[3] Another area that could be improved is the treatment of trust in modelling. The authors clarify the distinction between trust in models themselves and trust among stakeholders who use these models. While this is a relevant point, the explanation, though strong in the response letter, needs to be made more accessible and clearly articulated in the manuscript. This would help avoid misinterpretation, particularly by non-specialist readers, who often are key stakeholders in co-production frameworks.

We thank the reviewer for his critical reassessment of this point. We will follow his advice and expand on the issue of trust in modelling in the text. We propose to rephrase the text (ll 343) as follows (suggested additions to the text in bold):

"A possible way to embed this in existing workflows is to build standardized and simplified data and model pipelines, which encapsulate some of the complexity and allow for the easy running of separate models addressing the same question. This allows comparing model outputs of different approaches, which can support building trust in the models to be fit for **the intended** purpose. **While it should be acknowledged that running different models does not inherently enhance trust in the models, in particular when outputs are highly uncertain and variable, using several distinct modelling approaches building up to shared evidence and a common understanding of the phenomenon of interest is one of the main ways for scientists to build trust in models that are only hardly verifiable (Taylor et al. 2012; Merz et al. 2024). For instance, climate models may yield very large variances in global temperature outputs for the same emission trajectories. Confidence in the climate models is established by pooling the information from several, distinct models. This confidence was built thanks to a large-scale interoperability effort led by the CMIP team. This type of multi-model angle is not only required on the physical modelling side, where it is already common throughout the entire chain of information, but ideally for all models to best characterize uncertainties. However, this often shows how large they are, and is often difficult for stakeholders to fully understand. More importantly, disclosure of uncertainties does not inherently increase trust and credibility in risk analyses (Doyle et al., 2019), and thus information about uncertainty should be embedded in the co-production process and tailored to the specific audience, considering their perspectives, technical knowledge and concerns (Merz et al. 2024).**

**Crucially, to bridge this gap, there arises a need to build trust among modelers, stakeholders and decision-makers. Effective communication plays a pivotal role in establishing trust in DRM and CCA decisions. The proposed knowledge co-production process, i.e. connecting modellers, data providers and end users, promotes discussions regarding different modelling approaches and explores user needs vis-à-vis available information in a non-hierarchical manner (see Daniels et al., 2020). This is expected to increase the usability and accessibility of information by clarifying potential errors, uncertainties and underpinning assumptions embedded in each model for users, to align available information with the needs of planners and decision-makers. In other words, the co-production process is an ongoing negotiation between needs and what models can provide, which, insofar as uncertainty is accounted for, also generates trust in data through continuous and transdisciplinary engagement with it (Daniels et al., 2020).**

~~Importantly, tThe co-production process connects modellers, data providers and end users, promotes discussions regarding different modelling approaches, and explores user needs in contrast to available information (Daniels et al., 2020). In this process, effective communication is essential in establishing trust in DRM and CCA decisions. Uncertainties embedded in underlying data and models can be considerable. For stakeholders, uncertainties are often difficult to understand. Therefore, information about uncertainty should be embedded in the co-production process and tailored to the specific audience and consider their perspectives, e.g. Merz et al. (2024) for the example of flood hazard and risk modelling."~~

[4] Another example is the authors' response to reviewer #1's concern about vague terms such as "creative and interdisciplinary approaches". Although they now offer concrete examples such as games and participatory methods, I'd suggest reviewing them to ensure they are seamlessly integrated into the manuscript text.

We have carefully checked the manuscript for the terms "creative approaches", "interdisciplinary approaches", "specific roles and capacities". These terms are only used in section 3.1 Sub-system interoperability (ll 360) where, in response to the previous review, we provide the additional examples for clarification. We therefore think, that they are smoothly integrated to the text. The additional reference now included (Cumiskey et al. 2025) gives more context on the importance of building creative capacity for knowledge co-production,

References:

Cumiskey, L., Parviainen, J., Bharwani, S., Ng, N., Bagli, S., Drews, M., ... & Håkansson, V. W. (2025). Capacity development for locally-led knowledge co-production processes in Real World Labs for managing climate and disaster risk. International Journal of Disaster Risk Reduction, 125, 105398. https://doi.org/10.1016/j.ijdrr.2025.105398

Doyle, E. E. H., Johnston, D. M., Smith, R., and Paton, D.: Communicating model uncertainty for natural hazards: A qualitative systematic thematic review, International Journal of Disaster Risk Reduction, 33, 449–476, https://doi.org/10.1016/j.ijdrr.2018.10.023, 2019.

Parviainen, J., Hochrainer-Stigler, S., Cumiskey, L., Bharwani, S., Schweizer, P.-J., Hofbauer, B., and Cubie, D.: The Risk-Tandem Framework: An iterative framework for combining risk governance and knowledge co-production toward integrated disaster risk management and climate change adaptation, International Journal of Disaster Risk Reduction, 116, 105070, https://doi.org/10.1016/j.ijdrr.2024.105070, 2025.

Taylor, K. E., Stouffer, R. J., and Meehl, G. A.: An Overview of CMIP5 and the Experiment Design, Bulletin of the American Meteorological Society, 93, 485–498, , 2012. https://doi.org/10.1175/BAMS-D-11-00094.1